# Theoretical and Experimental Investigation of a Rotational Magnetic Couple Piezoelectric Energy Harvester

**DOI:** 10.3390/mi13060936

**Published:** 2022-06-12

**Authors:** Feng Sun, Runhong Dong, Ran Zhou, Fangchao Xu, Xutao Mei

**Affiliations:** 1School of Mechanical Engineering, Shenyang University of Technology, Shenyang 110870, China; sunfeng@sut.edu.cn (F.S.); 18843146714@163.com (R.D.); xufangchao@sut.edu.cn (F.X.); 2Institute of Industrial Science, The University of Tokyo, Tokyo 153-8505, Japan

**Keywords:** piezoelectric energy harvesting, frequency up-conversion, non-linear magnetic coupling, rotational motion

## Abstract

With the rapid development of Internet of Things (IoT) and the popularity of wireless sensors, using internal permanent or rechargeable batteries as a power source will face a higher maintenance workload. Therefore, self-powered wireless sensors through environmental energy harvesting are becoming an important development trend. Among the many studies of energy harvesting, the research on rotational energy harvesting still has many shortcomings, such as rarely working effectively under low-frequency rotational motion or working in a narrow frequency band. In this article, a rotational magnetic couple piezoelectric energy harvester is proposed. Under the low-frequency excitation (<10 Hz) condition, the harvester can convert low-frequency rotational into high-frequency vibrational of the piezoelectric beam by frequency up-conversion, effectively increasing the working bandwidth (0.5–16 Hz) and improving the efficiency of low-speed rotational energy harvesting. In addition, when the excitation frequency is too high (>16 Hz), it can solve the condition that the piezoelectric beam cannot respond in time by frequency down-conversion. Therefore, the energy harvester still has a certain degree of energy harvesting ability (18–22 Hz and 29–31 Hz) under high-frequency conditions. Meanwhile, corresponding theoretical analyses and experimental verifications were carried out to investigate the dynamic characteristics of the harvester with different excitation and installation directions. The experimental results illustrate that the proposed energy harvester has a wider working bandwidth benefiting from the frequency up-conversion mechanism and frequency down-conversion mechanism. In addition, the forward beam will have a wider bandwidth than the inverse beam due to the softening effect. In addition, the maximum powers of the forward and inverse beams at 310 rpm (15.5 Hz) are 93.8 μW and 58.5 μW, respectively. The maximum powers of the two beams at 420 rpm (21 Hz) reached 177 μW and 85.2 μW, respectively. The self-powered requirement of micromechanical systems can be achieved. Furthermore, this study provides the theoretical and experimental basis for rotational energy harvesting.

## 1. Introduction

With the advancement of wireless technology and microelectronics in recent years, allowing great development of IoT, the power supply issue has become one of the core issues of wireless sensor networks. For many devices installed in remote or not easily replaceable locations (such as IoT, bio-implantable devices, extremely harsh environments, etc.), self-powering of wireless sensing devices is extremely important. The traditional chemical batteries have problems such as the need to be replaced regularly, low energy density, and environmental pollution. To solve these existing problems, people started to harvest energy from the working environment of these low-power sensor devices. Piezoelectric materials have the advantages of directly generating available voltage, simple structure, small mechanical damping coefficient, high energy conversion efficiency, ease to achieve miniaturization and integration, etc. Therefore, piezoelectric energy harvesting has been studied by many scholars [1,2]. For example, Moradian et al. proposed a battery-less wireless Micro-Electro-Mechanical (MEMS)-based respiration sensor that can be applied in the medical field [3]. It is more convenient to monitor the patient’s breathing status over a long time period. Han et al. used beams made of alloy layers with different expansion coefficients [4]. After absorbing heat, the alloy beams deformed to counteract the magnetic force and caused the piezoelectric beam to vibrate. The thermal harvesting scheme shows application potential in heat event-driven autonomous monitoring. In addition, the structure of the energy harvester has been continuously improved [5] to increase energy harvesting efficiency and working bandwidth, which includes up-conversion [6,7,8], nonlinear [9,10,11], multimodal [12,13,14], frequency tuning [15,16], and so on.

Among the studies on piezoelectric energy harvesting, the recovery of vibration energy has been widely studied [17,18,19], because vibration energy is one of the most common energy sources in the environment, and the movement of objects is often accompanied by vibrations. In addition, energy harvesting from rotational motion should also be considered, as it can be used for condition monitoring of various rotating devices, for wind energy harvesting, etc. This type of energy harvester is usually composed of a rotor and a vibrator. The rotor is equipped with magnets. The vibrator is composed of a piezoelectric beam with a magnet at its free end. The rotor can be connected to external rotating equipment or rotating blades in the flow field and the magnet is used as an intermediate drive to pluck the piezoelectric beam and convert the vibration energy into electrical energy, which can be used to complete the frequency up-conversion and increase the working frequency bandwidth and the energy harvesting efficiency. For example, Fu et al. integrated up-conversion and a bistable mechanism by introducing two external input magnets [20]. Furthermore, this harvester has a wide bandwidth at low speed and achieves an asymmetric potential well by adjusting the magnetic potential. Therefore, the harvester has a stable and enhanced output and good control behavior. Hu et al. proposed an energy recycler made of U-shaped bimorph values; the electrical power generated by the harvester can reach 8.19 KW, which is comparable to the traditional small windmill generator, so the proposed harvester has attractive application prospects [21]. Xue et al. proposed three configurations to achieve out-of-plane plucking: direct repulsion configuration, orthogonal configuration, and indirect repulsion configuration. It is demonstrated that varying the number and spacing of beams and magnets enables an eccentric rotor-based up-conversion energy harvesting system to achieve advanced optimization [22].

In addition to plucking the beam with a magnet to make the piezoelectric beam vibrate, other structures can be used to collect the energy of the direct or indirect rotational energy. For example, Egbe et al. invented a vibrating turbine piezoelectric nanogenerator for collecting energy in a multiphase flow field [23]. Magnetic coupling is used to periodically deform the silicone rubber strips embedded in a piezoelectric film, thus effectively converting the vibration energy into electrical energy. Each bent piezoelectric silicone rubber strip can generate 9 μW of electrical power at 4 Hz. Zhao et al. studied a waterproof magnetically coupled piezoelectric-electromagnetic hybrid wind energy collector [24]. The piezoelectric ceramics are sandwiched between two flexural tension metal layers, and the force transfer of the flexible tension structure, pressure, and an amplified tension force perpendicular to the direction of pressure are simultaneously applied to both ends of the piezoelectric layers, which can eventually generate 1200 μW or so. Xie et al. proposed a rotating energy harvester with a piezoelectric bending beam with a central magnet and a pair of rotating magnets with opposite poles mounted on a rotating host to obtain low-speed rotational energy, which can steadily produce an output power of 6.91–48.01 µW in the rotational frequency range of 1–14 Hz. In addition, the harvester uses dual attraction magnets to overcome the suppression phenomenon at higher frequencies and can produce 42.19–65.44 µW in the range of 6–14 Hz. It also shows that the rotational motion energy harvester has great potential for environmental monitorin, and rotating machinery condition monitoring [25].

Most rotational energy harvesting currently requires higher excitation frequencies to work effectively and has a narrow working bandwidth. There are fewer studies for low-frequency conditions, so to solve these problems, a rotational magnetic couple piezoelectric energy harvester is proposed in this article, which consists of a turntable with a magnet in the middle and piezoelectric beams with tip magnets distributed in the upper and lower parts. Drive magnets rotate with the disc, and the piezoelectric beam with tip magnets is plucked by non-linear magnetic coupling, which in turn converts the vibration energy of the piezoelectric beam into electrical energy. The dynamic characteristics and harvesting capability of the harvester are theoretically analyzed and experimentally verified. The harvester can effectively achieve frequency up-conversion and increased energy harvesting efficiency at low-frequency. Meanwhile, the piezoelectric beam adopts two types of installation, inverse and forward, and it has multiple resonant rotational speeds. Thus, a rotating energy harvester has a wider working bandwidth and can realize energy harvesting in various environments.

The article is organized as follows. In Section 2, the structure and the principle of the rotational magnetic couple piezoelectric energy harvester are proposed and introduced briefly. The mathematical model of the harvester is derived in Section 3, and detailed experiments are conducted in Section 4 to discuss the effects on the harvester characteristics with different parameters. Section 5 is a parametric study in which the mathematical model is simulated and, finally, Section 6 is the conclusion.

## 2. Harvester Design and Operating Principle

Figure 1 shows the schematic diagram of the rotational magnetic couple piezoelectric energy harvester in this article. As shown in Figure 1a, the rotational energy harvester is divided into two parts, inverse cantilever beams and forward cantilever beams of 304 h stainless steel with a piezoelectric patch glued to the root of the beam and fixed to the upper and lower discs, respectively. A tip magnet is mounted at the free end of each cantilever beam by two L-plates, which correspond to the magnets on the rotating disc, and the same magnetic poles are close to each other. The rotation provided by the servo motor is to simulate the conditions of the direct connection of the energy harvester to the rotating machinery or the rotation provided by the blades rotating in the flow field. During the working process, the magnets on the rotating disc and the tip magnet of the beam are coupled periodically, and the repulsive force generated by the same magnetic pole will force the beam to bend, and as the driving magnet continues to rotate away from the tip magnet, the beam will produce damped-free-vibration under its elastic force before the next driving magnet comes. At the same time, the vibration energy is converted to electrical energy by piezoelectric energy. At this moment, frequency up-conversion is completed, and the low rotation frequency is converted into the higher vibration frequency of the beam so that the harvester can have a high energy harvesting efficiency even at a rotation frequency far below the resonance frequency of the beam. Then, the working bandwidth is improved.

Figure 1b shows a detailed diagram of the magnetic plucking process of the harvester, where *G* is the gravitational force, and $F_{m1}$ and $F_{m1}^{\prime }$ are the periodic non-linear magnetic forces generated by the driving magnet on the tip magnet, and its normal component makes the beam vibrate. Compared with the previous toggle and impact type, the magnetic plucking type has higher safety, longer service life, and less noise; the magnitude of excitation is also easily adjusted by the magnitude of magnetic force. Moreover, piezoelectric beams have several rotational speeds that can resonate, which can maintain high energy harvesting efficiency in a wider frequency range.

## 3. Theoretical Modeling and Analysis

To better analyze the dynamics of the harvester, a theoretical model needs to be built and analyzed; Figure 1b shows the configuration and structural parameters of one of the beams of the harvester. Using Hamilton’s principle, the control equation can be obtained:(1)$$M\ddot{x}+C\dot{x}+\left(K-K_{g}\right)x-\vartheta_{p}v_{1}-F_{m1}-G_{1}=0$$
(2)$$M\ddot{x}+C\dot{x}+\left(K+K_{g}^{\prime }\right)x-\vartheta_{p}v_{2}-F_{m1}^{\prime }+G_{1}=0$$
(3)$$C_{p}{\dot{v}}_{1}+R_{l}^{-1}v_{1}+\vartheta_{p}\dot{x}=0$$
(4)$$C_{p}{\dot{v}}_{2}+R_{l}^{-1}v_{2}+\vartheta_{p}\dot{x}=0$$
where M is the equivalent mass, C is the mechanical damping coefficient, K is the equivalent stiffness of the piezoelectric beam, and $K_{g}$ and $K_{g}^{\prime }$ are the equivalent stiffnesses generated by the gravity component along the forward and inverse beam direction, respectively. $\vartheta_{p}$ is the electromechanical coupling coefficient of the piezoelectric patch, $v_{1}$ and $v_{2}$ are the output voltages of the forward and inverse beams, $F_{m1}$ and $F_{m1}^{\prime }$ are the non-linear magnetic force along the transverse deformation direction for both piezoelectric beams, $G_{1}$ is the component of gravity perpendicular to the beam, $C_{p}$ is the equivalent capacitance, and $R_{l}$ is the electrical load. As shown in Figure 2a, we mainly study the transverse displacement of the piezoelectric beam in the process of magnetic plucking, whereas the mass of the tip magnet is much larger than the mass of the piezoelectric beam, so the first mode dominates in the Galerkin expansion [26]; then, the transverse displacement can be written as:(5)w(x,t)=ψ(x)q(t)
where ψ(x) is the first-order vibration of the cantilever beam, q(t) is the generalized temporal displacement, and the tip displacement of the beam can then be expressed as w(L,t)=ψ(L)q(t).

When the cantilever beam vibrates at the same time, the tip magnet will also rotate with an angle θ, as shown in Figure 2b. This angle can be approximated by the horizontal displacement of the beam as $\theta =\arcsin\left(w^{\prime }\left(L,t\right)\right)$. The vertical distance between magnet A and magnet B is *d* and the horizontal distance is *s*, which can be expressed as:(6)$$\Delta y=\frac{t_{A}}{2}-\frac{t_{A}\cos\theta }{2}+L+\frac{t_{A}}{2}-\sqrt{{\left(L+\frac{t_{A}}{2}\right)}^{2}-w^{2}\left(L,t\right)}$$
(7)$$d=d_{0}+\frac{t_{A}}{2}-\frac{t_{A}\cos\theta }{2}+L+\frac{t_{A}}{2}-\sqrt{{\left(L+\frac{t_{A}}{2}\right)}^{2}-w^{2}\left(L,t\right)}$$
(8)$$S=-h+w\left(L,t\right)+\frac{t_{A}\sin\theta }{2}$$
where Δy represents the vertical deformation due to the horizontal deformation of the cantilever beam, $t_{A}$ represents the thickness of the tip magnet A, and *h* (h=Rsinωt) is the horizontal distance between the magnet B and the magnet A at w(L,t)=0. The distance between magnets perpendicular to the vibration direction of the beam is so small that it is negligible. *R* is the radius of rotation of the magnet B, ω is the angular velocity of rotation. For tip magnet A and rotating magnet B, the magnetic moment vector can be expressed as:(9)$$\mu_{A}=M_{A}V_{A}\sin\theta {\hat{e}}_{x}+M_{A}V_{A}\cos\theta {\hat{e}}_{y}$$
(10)$$\mu_{B}=-M_{B}V_{B}{\hat{e}}_{y}$$
where $M_{A}$ and $M_{B}$ are the magnetization intensity of magnet A and magnet B, $V_{A}$, $V_{B}$ is the volume of magnet A and magnet B, and ${\hat{e}}_{x}$, ${\hat{e}}_{y}$ is the unit vector along the X-axis Y-axis. The direction vector $r_{\mathrm{BA}}$ from $\mu_{B}$ to $\mu_{A}$, according to the geometric relationship in Figure 2b, can be written as:(11)$$r_{BA}=s{\hat{e}}_{x}-d{\hat{e}}_{y}$$

For the calculation of the nonlinear magnetic force, using the magnetic dipole method, the magnetic induction intensity at magnet A is expressed as:(12)$$B_{BA}=-\frac{\mu_{0}}{4\pi }𝛻\frac{\mu_{B}\cdot r_{BA}}{∥r_{BA}∥{}_{2}^{3}}$$

The potential energy at the magnet A can then be written:(13)$$U_{BA}=-B_{BA}\cdot \mu_{A}$$
(14)$$U_{BA}=\frac{\mu_{0}}{4\pi }𝛻\left(\frac{\mu_{B}\cdot r_{BA}}{∥r_{BA}∥{}_{2}^{3}}\right)\mu_{A}=\frac{\mu_{0}}{4\pi }\left(\frac{\mu_{B}}{∥r_{BA}∥{}_{2}^{3}}-\left(\mu_{B}\cdot r_{BA}\right)\frac{3r_{BA}}{∥r_{BA}∥{}_{2}^{5}}\right)\mu_{A}$$
where is the $\mu_{0}$ magnetic permeability constant, 𝛻 is the vector gradient operator, $\mu_{A}$ and $\mu_{B}$ are the magnetic moments of magnets A and B, respectively, and $∥{∥}_{2}$ denotes the Euclidean norm. The magnetic force can be obtained by taking the derivative of Equation (14) with respect to $r_{\mathrm{BA}}$, as follows:(15)$$F_{m}=-𝛻U_{BA}=\frac{3M_{A}V_{A}M_{B}V_{B}\mu_{0}\left[\left({\hat{\mu }}_{A}\cdot {\hat{\mu }}_{B}\right){\hat{r}}_{BA}+\left({\hat{\mu }}_{B}\cdot {\hat{r}}_{BA}\right){\hat{\mu }}_{A}+\left({\hat{\mu }}_{A}\cdot {\hat{r}}_{BA}\right){\hat{\mu }}_{B}-5\left({\hat{\mu }}_{A}\cdot {\hat{r}}_{BA}\right)\left({\hat{\mu }}_{B}\cdot {\hat{r}}_{BA}\right){\hat{r}}_{BA}\right]}{4\pi ∥r_{BA}∥{}_{2}^{4}}$$
where ${\hat{\mu }}_{A}$, ${\hat{\mu }}_{B}$, and ${\hat{r}}_{BA}$ are the unit vectors along $\mu_{A}$, $\mu_{B}$, and $r_{BA}$, respectively. It follows that the magnetic force is divided in the x-direction as follows:(16)$$F_{m1}=\frac{-3M_{A}V_{A}M_{B}V_{B}\mu_{0}\left[\left(dsin\theta -scos\theta \right)\left(d^{2}+s^{2}\right)+5ds\left(dcos\theta -ssin\theta \right)\right]}{4\pi \left(d^{2}+s^{2}\right){\;}^{\frac{7}{2}}}$$

In addition, to rate the energy harvesting capability of the harvester, the root-mean-square voltage and the average power are used, as follows:(17)$$v_{rms}=\sqrt{\frac{1}{T}\int_{0}^{T}v{\left(t\right)}^{2}dt}$$
(18)$$P_{avg}=\frac{v_{rms}^{2}}{R_{l}}$$

## 4. Experimental Validations

### 4.1. Experimental Setup

To study the dynamic characteristics and energy harvesting performance of the proposed harvester, an experimental setup is shown in Figure 3. It was built and the lateral displacement and voltage output of the setup was tested at different rotational speeds. For more accurate control of the rotational speed, the rotation of the turntable was provided by a servo motor (SGM7J Yaskawa, Kitakyushu, Japan), and the parameters of rotary motion are set by Programmable Automation Controller (PAC), and then the matching servo actuator (SGD7S, Yaskawa) can control the motion of the servo motor. The purpose is to simulate the rotational force provided by the direct connection to the rotating equipment or the blades in the flow field. Three uniformly distributed cylindrical NdFeB magnets are glued to each of the top and bottom sides of the turntable, and the magnets have a size of D12 × 2 mm³ (D is the diameter). In addition, laser displacement sensors (HG-C1050, HG-C1100, Panasonic, Osaka, Japan) are used to measure the lateral displacement of the beams (not at the tip mass). Each beam has a tip magnet at the free end, again a cylindrical NdFeB magnet of the same size. In addition, the fixed end of the beam is glued with a lead zirconate-titanate (PZT) piezoelectric ceramic (K2512U1, Thrive, Tokyo, Japan) on one side, and the length of the piezoelectric beam is 90 mm. An oscilloscope (MDO-2204ES Gwinstek, New Taipei, Taiwan) was used to monitor and record the displacement and output voltage of the harvester, which was used to analyze the amplitude-frequency characteristics and energy harvesting efficiency of the piezoelectric beam.

In addition, different experiments were conducted for the spacing between the rotating magnets and the tip magnets to analyze the dynamics and energy harvesting capability of piezoelectric beams under different excitation magnitudes.

### 4.2. Experimental Results and Discussion

The experimental results of the RMS voltage of inverse beam with 10 mm magnet pitch are shown in Figure 4. The experimental rotational speeds were from 10 rpm (0.5 Hz) to 700 rpm (35 Hz). In region I, the excitation frequency is lower currently. At the speed of 90 rpm, the excitation frequency is 4.5 Hz. However, as shown in Figure 5d, the power spectrum is calculated by the Fast Fourier Transform (FFT); it can be noticed that the frequency of the output voltage is 11 Hz, which is higher than the excitation frequency. The reason is that the natural frequency of the piezoelectric beam is greater than the excitation frequency at this time, so a special behavior of the harvester emerged is the ring-down pattern. Before the next magnetic coupling between the driving magnet and the tip magnet, the piezoelectric beam can still be vibrating from the preceding plucking and is in damped-free-vibration form. This mechanism allows the energy harvester to operate at its resonant frequency when low-frequency motion is called frequency up-conversion. The low excitation frequency is converted to the high vibration frequency of the piezoelectric beam. So, the harvester can have high-efficiency energy harvesting at a low-frequency. Furthermore, it is obvious in Figure 5a that red squares are not typical damped-free-vibrations because the vibration of the piezoelectric beam due to the previous excitation does not stop when the next excitation arrives. When the phases of both excitation frequency and vibration frequency of the piezoelectric beam are the same, the amplitude of the piezoelectric beam will be increased, as shown in Figure 5(a1); if their phases are not the same, the amplitude will be weakened, as shown in Figure 5(a2). When the rotational speed reaches 200 rpm (10 Hz), according to Figure 5h, the piezoelectric beam generates the main harmonic resonance. This is because excitation frequency is close to the natural frequency of the piezoelectric beam. The frequency of Figure 5h is less than 10 Hz, caused by the lack of control accuracy of PAC, but it does not affect the mechanical characteristics generated by the piezoelectric beam.

When reaching region II, the excitation frequency exceeds the intrinsic frequency of the piezoelectric beam, as Figure 6d at 270 rpm (13.5 Hz) shows. Due to the different operating frequencies of the vibrational beam and the driving magnets, the phase difference between the magnetic force and the velocity of the beam tip is caused. Because of their joint influence, super-harmonic resonance appears at this moment, and the phase mismatch between the excitation force and the beam vibration will be more serious in this region. When the rotational speed continues to increase to 330 rpm (16.5 Hz) above, which is region III in Figure 4, the voltage output will suddenly drop to about 2 V. Because the excitation time from the magnetic coupling becomes shorter and shorter as the rotational speed continues to increase until it is less than the response time of the piezoelectric beam, the driving magnet is not able to pluck the piezoelectric beam effectively and the piezoelectric beam will only produce a transient response. Therefore, this phenomenon leads to a smaller amplitude of the piezoelectric beam, which reduces the output voltage and finally makes the energy harvesting capability drop sharply.

However, at high-frequency excitation, the piezoelectric beam will not always be at low amplitude. Figure 4’s black dotted square shows that when the speed is increased near 400 rpm (20 Hz) and 600 rpm (30 Hz), the RMS voltage appears to re-rise. The excitation frequency at this moment is exactly two and three times the natural frequency of the piezoelectric beam, but the piezoelectric beam resonates at its natural frequency (as shown in Figure 7d,h). At this point, a frequency down-conversion mechanism appears, benefiting from which the amplitude increases again and the RMS voltage rises. As a result, the harvester designed in this article has a wider bandwidth, which is different from the case of a simple harmonic piezoelectric vibratory harvester, which has only one optimal frequency and a narrow bandwidth.

When changing the magnitude of the magnetic force, the dynamic characteristics of the piezoelectric beam are consistent and can be seen in Figure 8. It is still divided into three regions. However, the difference is that the decrease in the magnet spacing, i.e., the increase in the magnetic force, will make the piezoelectric beam respond more rapidly, and the beam will enter the mode indicated by region III later. Furthermore, as shown in the black dotted square in the figure, the low-energy-harvesting region will be a little smaller at the 8 mm spacing. Conversely, this means that the increase in excitation will result in a wider bandwidth.

The energy harvester is divided into upper and lower parts, with the piezoelectric beam installed inverse and forward, respectively, and in Figure 9 their RMS voltages can still be divided into three regions with similar trends. However, the black dotted square in the figure shows a difference in the low-energy-harvesting region. According to the difference in mounting, it can be known that the two piezoelectric beams are subjected to magnetic forces in opposite directions at the tips and gravity is vertically downward, which causes the combined forces on the tips to be different. As shown in Figure 10a, because the component force of gravity and the component force of magnetism are in opposite directions, the force perpendicular to the inverse beam is reduced. At the same time, gravity also gives the beam a hardening effect, which is not helpful for the vibration of the beam. However, the reverse is true for the forward beam, where the partition of gravity not only softens the beam but also increases the force perpendicular to the beam, as shown in Figure 10b. This therefore creates a difference in the low-energy-harvesting region and gives the forward beam a wider bandwidth. This difference is consistent with the phenomenon produced by changing the magnitude of the magnetic force and justifies the correctness of the above point.

Under real energy harvesting performance, it is closely related to the load resistance connected to the piezoelectric patches. To test the effect of load resistance on the output power, Figure 11 shows the average power of the inverse beam and forward beam at different load resistances at 10 mm magnet spacing. Initially, the output power increases with the increasing resistance until it approaches 150 KΩ. At 420 rpm (21 Hz), the output power of the forward beam even reaches 177 μW and the maximum power of the inverse beam is 85.2 μW. Specifically, this result is obtained because the piezoelectric beam resonates around its natural frequency of about 10 Hz due to the down-conversion mechanism. Meanwhile, the two beams also reached their maximum output power at 310 rpm (15.5 Hz) with 93.8 μW and 58.5 μW, respectively. However, with a further increase in the load resistance, the output power instead continues to decrease. The experimental results prove that the optimal load resistance of the energy harvester is about 150 KΩ. Meanwhile, the resistance that maximizes the power can be calculated from the following equation [27]:(19)$$R=\frac{1}{2\pi C_{p}f}$$
where $C_{p}$ of the capacitance of the piezoelectric patch is 115 nF and f is the frequency of the voltage of about 10 Hz. The result of the calculation is approximately 138 KΩ. The calculated results are in general agreement with the experimental results.

In particular, the experiments use a resistor to simulate the input resistance of the harvesting circuit. In practice, the energy collected by the energy harvester needs to be stored in the capacitor in order to power the sensor. For example, the rotating axis of the energy harvester is connected to the external rotating equipment to realize long-term uninterrupted monitoring of the working equipment. Alternatively, the energy harvester can be equipped with blades and placed in the flow fields, such as wind and water, to complete long-term autonomous environmental monitoring and realize the self-power supply of the system.

## 5. Parametric Studies

Finally, to gain insight into the characteristics of the rotating energy harvester, different speeds and configurations were explored through numerical parameter studies. For the numerical simulations, Equations (1)–(4) were solved in MATLAB (R2017a, MathWorks, Natick, MA, USA) software using the Runge–Kutta Method. The initial parameters used for the numerical simulations are shown in Table 1. The effect of these parameters on the energy harvesting performance under different conditions is investigated. It is important to note that in the following parametric studies when changing one parameter, the other parameters are the same as the initial parameters. Figure 12 shows the simulation results of the RMS voltage at different speeds for magnet spacing of 8 mm and 10 mm, respectively. The phenomenon in the dotted square shows that it is consistent with the results obtained in the experimental section. That is, the low capture energy region decreases as the magnet gap decreases (i.e., the excitation increases), and this means that a larger excitation is accompanied by a wider working bandwidth. The RMS voltage also increases due to the reduction in the gap.

Figure 13 and Figure 14 show the displacement, voltage, and spectrum at 90 rpm (4.5 Hz) and 400 rpm (20 Hz), respectively. The simulation results also show both up-conversion and down-conversion mechanisms of the energy harvester. This gives it a good energy harvesting efficiency at low frequencies while maintaining a certain energy harvesting capability at high excitation frequencies. Similarly, the simulations in Figure 15 for the forward and inverse beams again explain the difference in the dynamic characteristics of the two beams in the experiment. The influence of the tip mass indirectly changes the magnitude of the excitation force and produces both softening and hardening effects. This makes the working bandwidth and energy harvesting efficiency of the forward beam always slightly larger than that of the inverse beam under the same conditions.

In the previous section, the dynamic characteristics of the energy harvester at different speeds and their mechanisms were discussed. To illustrate it more intuitively from theory, numerical simulations and experimental results are compared in Figure 16. In the speed range of 10–310 rpm, the experimental test results are in better agreement with the numerical simulation results. At 320 rpm, the reduction in energy harvesting efficiency due to the inability of the piezoelectric beam to respond in time was predicted. In addition, at 400 rpm, the resonance-induced frequency down-conversion mechanism was successfully predicted and the RMS voltage was re-escalated. The discrepancy around 600 rpm may be because the piezoelectric beam in the experiment is affected by the tip gravity and cannot guarantee absolute verticality and equilibrium, which makes the experimental piezoelectric beam more susceptible to receiving the effect of the down-conversion mechanism.

The parametric study in this section provides the theoretical basis and theoretical guidance for the subsequent rotational magnetic couple piezoelectric energy harvesting.

## 6. Conclusions

This article proposes a rotational magnetic couple piezoelectric energy harvester that can be used to harvest rotating energy in various environments. Meanwhile, the theoretical model is derived to analyze the effects of non-linear magnetic force, softening and hardening on the output voltage. Additionally, comparative experiments were conducted to investigate the dynamics of the piezoelectric beam with different configurations and rotational speeds to validate its performance in energy harvesting. The following conclusions can be obtained:(1)The proposed energy harvester has a wider working bandwidth benefiting from the frequency up-conversion mechanism at low-frequency conditions and the frequency down-conversion mechanism at high-frequency.(2)The gap distance of interaction magnets decreasing and the softening effect produced by the forward beam both lead to a wider bandwidth of the energy harvester.(3)The optimal load resistance for 10 mm pitch is about 150 KΩ. The maximum average power of the forward and inverse beams at 310 rpm (15.5 Hz) was 93.8 μW and 58.5 μW, respectively. At 420 rpm (21 Hz), the maximum average power of the two beams can reach 177 μW and 85.2 μW, respectively.

In summary, the proposed harvester has a wide operating bandwidth and good energy harvesting efficiency under low-frequency excitation and high-frequency excitation. It can generate enough output power to power the wireless sensors, which can be applied in the fields of environmental monitoring and rotating machinery monitoring with high safety and reliability. In addition, this research provides a theoretical basis and experimental guidance for rotational energy harvesting. Future research will focus on changing the stiffness of the cantilever beam to achieve a further widening of the rotating energy harvester bandwidth.

## Figures and Tables

**Figure 1 micromachines-13-00936-f001:**
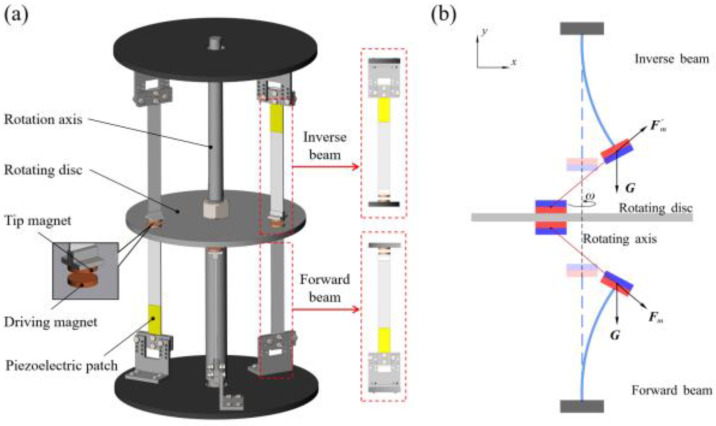
(**a**) Schematic diagram of rotational magnetic couple piezoelectric energy harvester; (**b**) schematic diagram of the magnetic plucking process of the harvester.

**Figure 2 micromachines-13-00936-f002:**
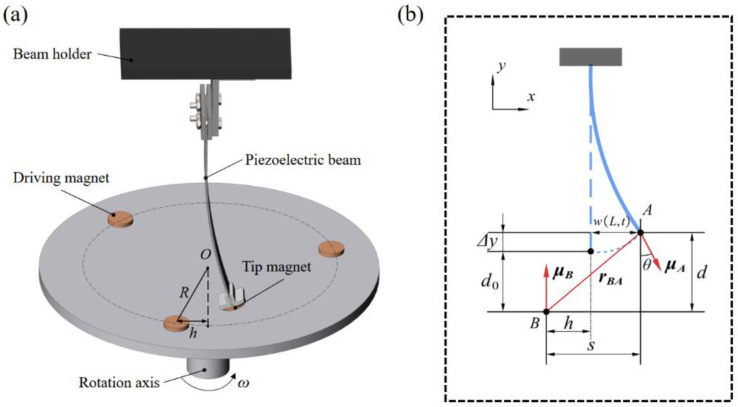
(**a**) Schematic diagram of the repulsive magnets; (**b**) the geometrical relations of magnets configuration via dipole-dipole methods.

**Figure 3 micromachines-13-00936-f003:**
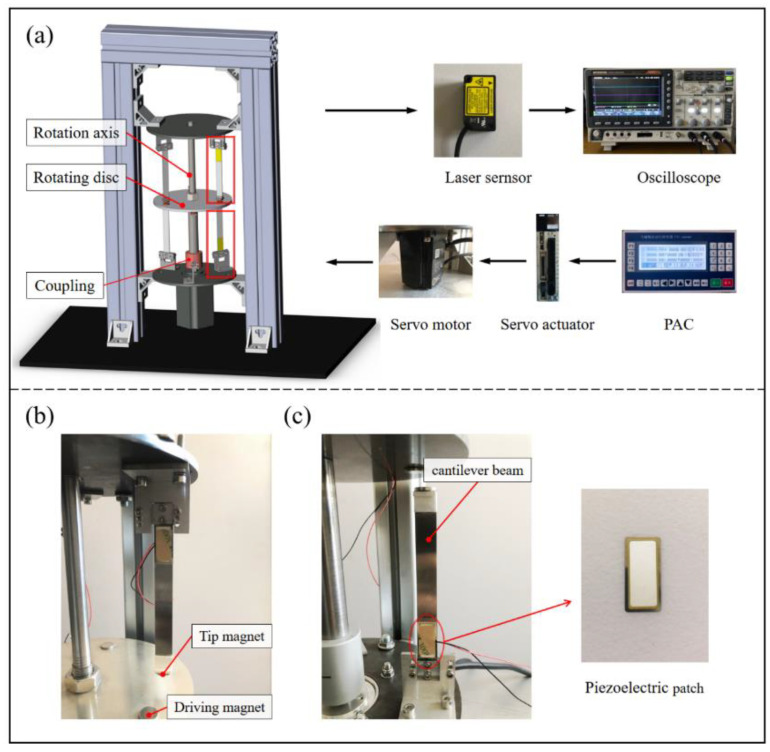
(**a**) Experimental setup of a rotational magnetic couple piezoelectric energy harvester, (**b**) the inverse piezoelectric beam, and (**c**) the forward piezoelectric beam.

**Figure 4 micromachines-13-00936-f004:**
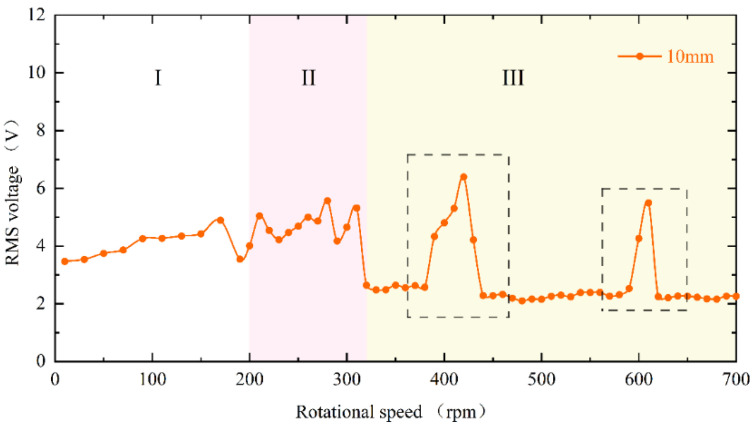
Experimental results of inverse beam RMS voltage at 10 mm magnet spacing.

**Figure 5 micromachines-13-00936-f005:**
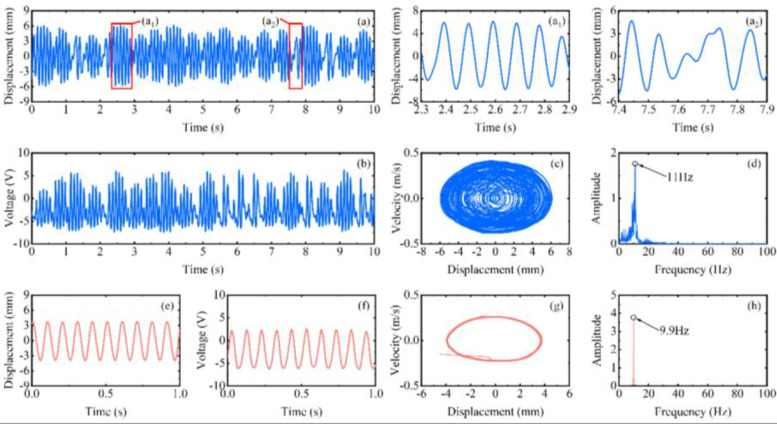
Experimental results of the inverse beam at 90 rpm (blue line) and 200 rpm (red line), (**a**,**e**) are the corresponding measured displacements, ((**a1**,**a2**) are enlarged view of the red square in (**a**)), (**b**,**f**) are the corresponding measured output voltages, (**c**,**g**) are the displacement-velocity phase portrait, and (**d**,**h**) are the corresponding power spectrums.

**Figure 6 micromachines-13-00936-f006:**
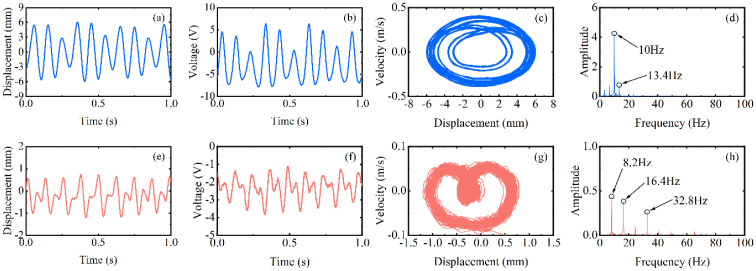
Experimental results of the inverse beam at 270 rpm (blue line) and 330 rpm (red line), (**a**,**e**) are the corresponding measured displacements, (**b**,**f**) are the corresponding measured output voltages, (**c**,**g**) are the displacement-velocity phase portrait, and (**d**,**h**) are the corresponding power spectrums.

**Figure 7 micromachines-13-00936-f007:**
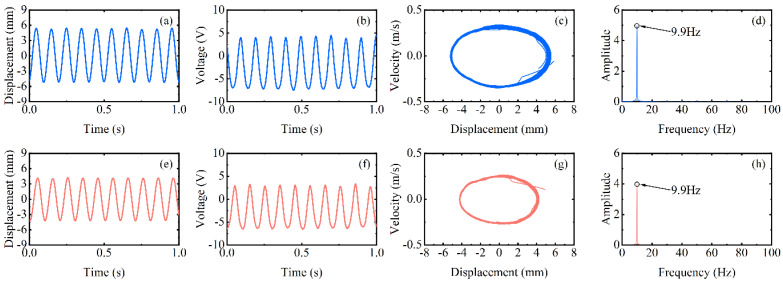
Experimental results of the inverse beam at 400 rpm (blue line) and 600 rpm (red line), (**a**,**e**) are the corresponding measured displacements, (**b**,**f**) are the corresponding measured output voltages, (**c**,**g**) are the displacement-velocity phase portrait, and (**d**,**h**) are the corresponding power spectrums.

**Figure 8 micromachines-13-00936-f008:**
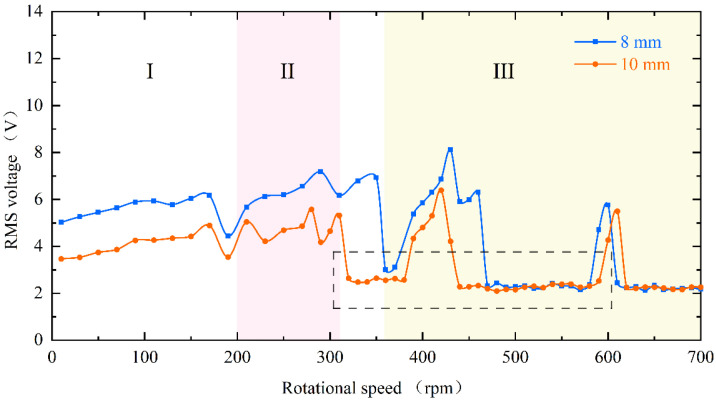
Experimental results of RMS voltage for the inverse beam at magnet spacing of 8 mm (blue line) and 10 mm (red line).

**Figure 9 micromachines-13-00936-f009:**
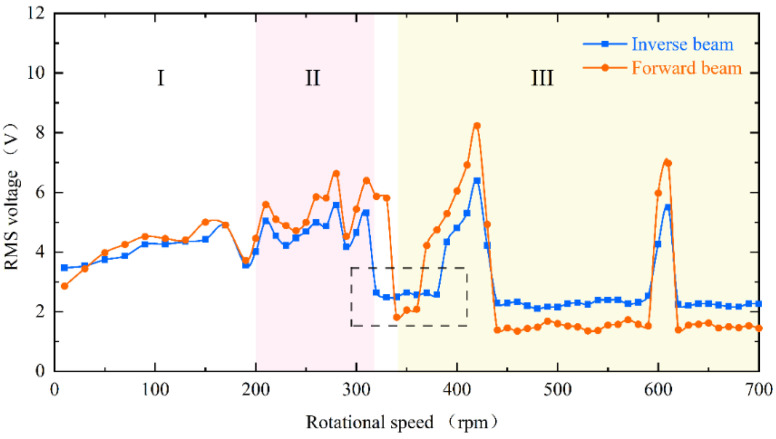
The experimental results of the RMS voltage comparison between the inverse beam (blue line) and forward beam (red line) at 10 mm magnet spacing.

**Figure 10 micromachines-13-00936-f010:**
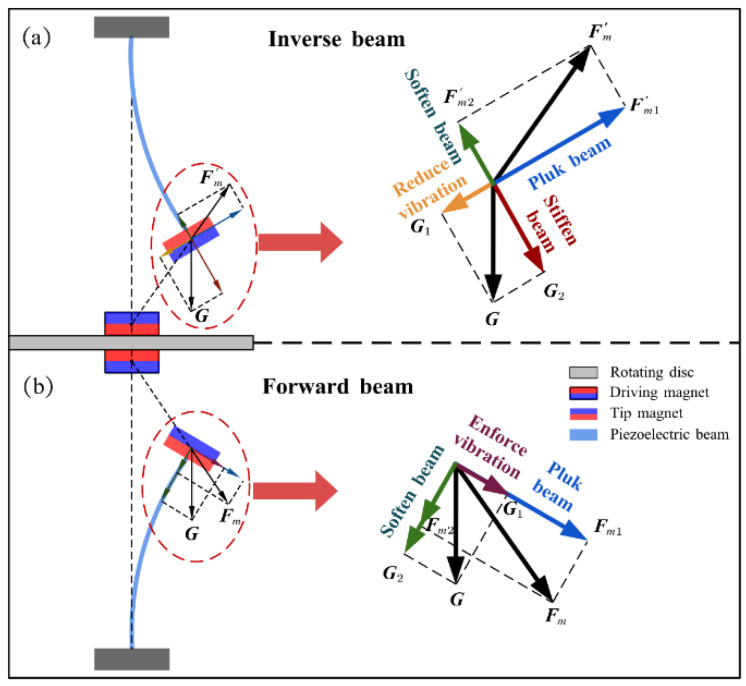
(**a**) Force analysis diagram of the tip of the inverse beam. (**b**) Force analysis diagram of the tip of the forward beam.

**Figure 11 micromachines-13-00936-f011:**
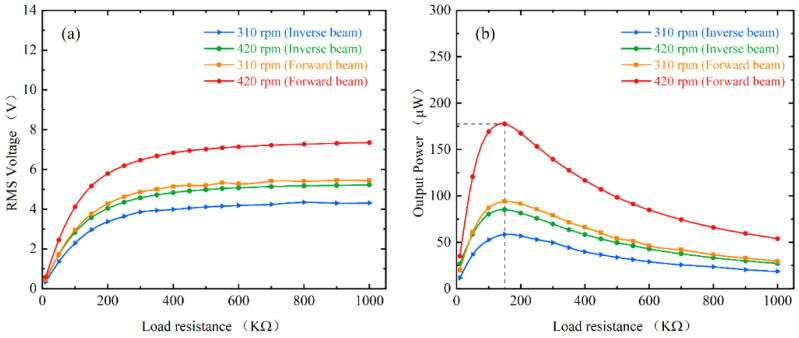
Experimental results of energy harvester under different load resistances: (**a**) RMS output voltage; (**b**) Average output power.

**Figure 12 micromachines-13-00936-f012:**
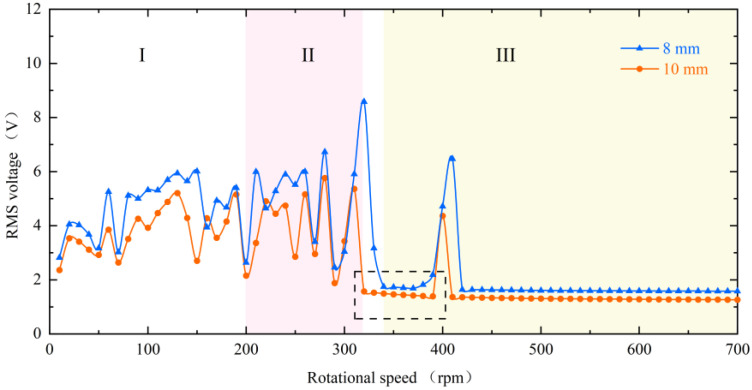
Simulation results of RMS voltage for the inverse beam at magnet spacing of 8 mm (blue line) and 10 mm (red line).

**Figure 13 micromachines-13-00936-f013:**
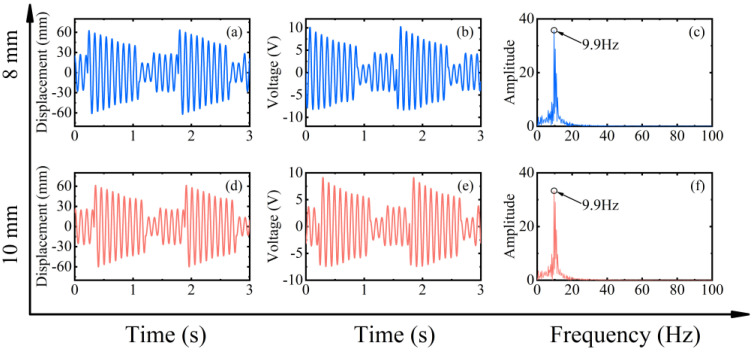
Simulation results of magnet spacing of 8 mm (blue line) and 10 mm (red line) at 90 rpm (4.5 Hz) in frequency up-conversion mechanism, (**a**,**d**) are the corresponding measured displacements, (**b**,**e**) are the corresponding measured output voltages, and (**c**,**f**) are the corresponding power spectrums.

**Figure 14 micromachines-13-00936-f014:**
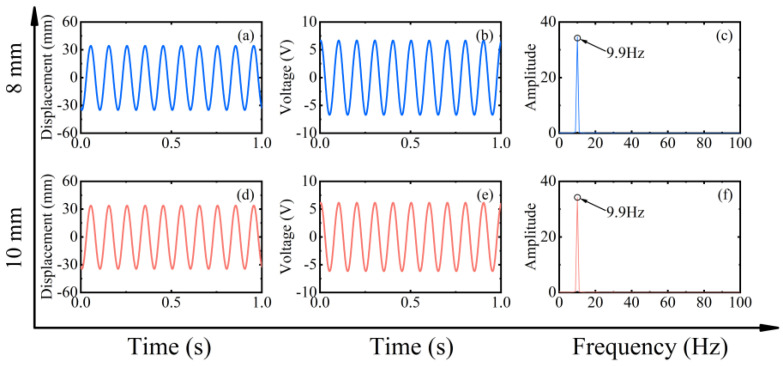
Simulation results of magnet spacing of 8 mm (blue line) and 10 mm (red line) at 400 rpm (20 Hz) in frequency down-conversion mechanism, (**a**,**d**) are the corresponding measured displacements, (**b**,**e**) are the corresponding measured output voltages, and (**c**,**f**) are the corresponding power spectrums.

**Figure 15 micromachines-13-00936-f015:**
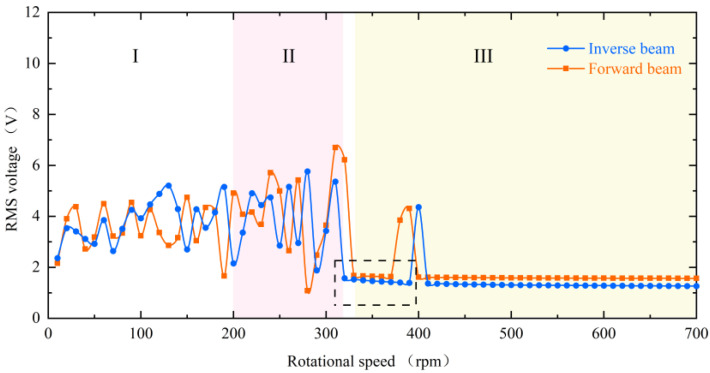
Simulation results of the RMS voltage comparison between the inverse beam (blue line) and forward beam (red line) at 10 mm magnet spacing.

**Figure 16 micromachines-13-00936-f016:**
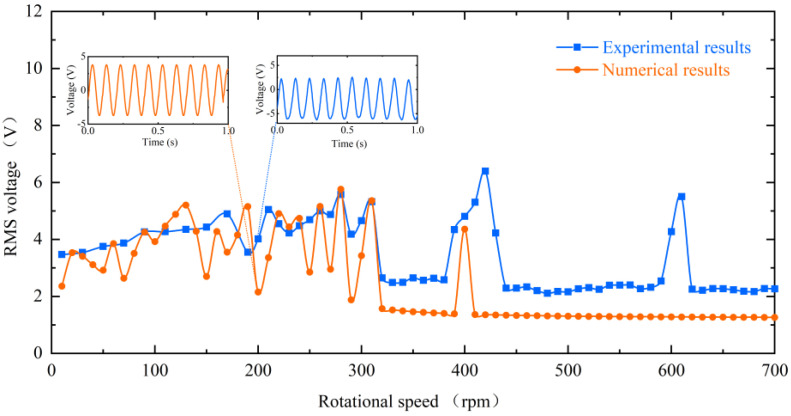
Comparison of simulation results and experimental results of RMS voltage for inverse beams at 10 mm magnet spacing.

**Table 1 micromachines-13-00936-t001:** Parameters of the piezoelectric energy harvester configuration.

Parameters	Symbol	Value
**Cantilever beam**		
Length × Width × Thickness	*L × b × t_b_*	90 × 12 × 0.25 mm^3^
Density	*ρ_b_*	7765 Kg/m^3^
Young’s modulus	*E_b_*	210 Gpa
Mechanical damping coefficient	*C*	0.005 N∙s/m
Equivalent stiffness	*K*	16.85 N/m
**Piezoelectric patch**		
Length × Width × Thickness	*L × b × t_p_*	25 × 12 × 0.13 mm^3^
Density	*ρ_p_*	7800 Kg/m^3^
Young’s modulus	*E_p_*	66 Gpa
Coupling coefficient	*d* _31_	−320 × 10^12^ C/N
Permittivity constant	*ε* _33_	4000 ε_0_
Permittivity of free space	*ε* _0_	8.854 × 10^12^ F/m
Electromechanical coupling coefficient	*ϑ_p_*	1.197 × 10^−5^
Equivalent capacitance	*C_p_*	1.15 × 10^−7^ F
**Permanent magnets**		
Density	*ρ_m_*	7500 Kg/m^3^
Magnet’s residual flux density	*B_r_*	1.5 T
Permeability of free space	*μ* _0_	4 π × 10^−7^
Magnet A	*D × t_m_*	D12 × 2 mm^3^
Magnet B	*D × t_m_*	D12 × 2 mm^3^
**Other parameters**		
Equivalent stiffness of gravity	*K_g_ K_g_^′^*	0.6 N/m
Whole load resistance	*R* _l_	1000 KΩ

## Data Availability

Data sharing not applicable to this article.

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
