# Peer review of "Theoretical and Experimental Investigation of a Rotational Magnetic Couple Piezoelectric Energy Harvester"

_micromachines, 2022, doi:10.3390/mi13060936_

Round 1

Reviewer 1 Report

Review of Theoretical and experimental investigation of a rotational magnetic couple piezoelectric energy harvester

Abstract: Good abstract, add details on working bandwidth, frequency ranges for up and down-conversion, and power outputs of each to engage the reader even more (see below for more detail).

Introduction:

Content is good, grammar improvements:

Line 34-37 needs to be re-written.

Lines 38-41 needs to be broken into two sentences.

Line 41- used "improved" twice

What is "so on" at end of line 43? Any other references?

Describe the system before explaining how it works in lines 49-52.

Break down into several sentences lines 52-61.

Is the sentence in lines 61-65 from a reference? Please indicate.

Use periods to separate sentences in the paragraph of lines 66-80. Include the citation number at end of line 85.

Section 2.

Indicate inverse and forward cantilever beams in Fig. 1 It is not clear.

What is a tip magnet? How are these magnets attached to the tip of the beam?

Section 4.

How can this system be used to harvest energy? The table is rotated with a motor in the current experiment. The authors should provide direct examples of where this energy harvester could be used. Most vibration is not rotational in nature.

What piezoelectric material is used?

Consistently leave a space between numbers and units, such as in “90 mm” or “11 Hz”.

What is a PAC? In figure 3.

The test results seem contradictory to what the abstract described. The most efficient operation is actually a result of down-conversion of frequency since the 172 µW is produced at 420 rpm (~20 Hz) while the beams are resonating at ~10Hz. Up-conversion is also observed but not for the most efficient results.  I think it would be beneficial to include both of these mechanisms in the results and in the abstract.

Up-conversion is still beneficial as low-frequency vibration can be harvested but with lower power efficiency. It would be useful to this publication to indicate what this efficiency is by measuring the power harvested as done in Figure 11 for the lower frequencies.

Also, make sure that it is stated that the max power (177 uW with a 150 Kohm load) is obtained as a down-conversion mechanism when the table is rotated at 20 Hz and the piezoelectric beams resonate at 10 Hz.

Also, please provide examples and a discussion of where this harvester can be used.

Conclusions

Include the conclusions of up and down-conversion and indicate what max power conversion is for the different frequency settings.

Reviewer 2 Report

The submitted manuscript titled “Theoretical and experimental investigation of a rotational magnetic couple piezoelectric energy harvester” describes the design of a rotational magnetic couple piezoelectric energy harvester. A theoretical analysis as well as experimental measurements are presented and illustrate the capability of the proposed vibration energy harvester in scavenging energy on a relatively large frequency band. The analysis part in the experimental section is particularly interesting. However, the manuscript presents major issues that need to be addressed before publication. Here are my comments:

Major comments

-I think the introduction has to be rewritten, for the following reasons:

  • In the first paragraph, the authors start introducing piezoelectric energy harvesting (“Among many energy harvesting techniques, piezoelectric materials have been studied”). In the next paragraph, the authors talk about vibration energy harvesting, which is a broader field than piezoelectric energy harvesting (“Among the studies on energy recovery, the recovery of vibration energy has been widely studied”). I think that the authors should start describing vibration energy harvesting before describing piezoelectric energy harvesting. Indeed, the introduction should start with a broad view of the field, and should get narrower.
  • I do not recommend the use of multiple references in a single sentence “[1]-[6]”, “[16]-[19]”, “[7]-[12]”. One or two references can be used to illustrate a point, but illustrating a single sentence with six references does not bring much value to the text. If the authors want to include a reference, they should write a single sentence summarizing the idea or main innovation of the reference, instead of adding six references in a single sentence without describing them.

-Equation (18) is not rms power, but average power. Rms power does not have any physical meaning.

Therefore Prms  should be renamed Pavg and this should also be corrected in the text of the new version of the manuscript.

Please do read the following note for more information:

https://www.analog.com/en/analog-dialogue/raqs/raq-issue-177.html

-There is no figure showing the results of the theoretical analysis. Furthermore, the results of the theoretical analysis are not compared with the experimental results. Therefore, the usefulness of the theoretical analysis remains extremely limited in the current state of the manuscript. This is a major issue of the manuscript. I advise the authors to:

  • Add a thorough analysis of the proposed model, and provide figures that show the theoretical influences of parameters on the behavior of the harvester (in order to obtain an analysis from their model).
  • Compare the experimental results with the theoretical predictions that can be made with the theoretical model.

-The authors claim that the bandwidth of their harvester is large (in the conclusion for example): “The proposed energy harvester has a wider working bandwidth benefiting from the frequency up-conversion mechanism caused by the rotational motion”.  However, I do not see any measurement of the bandwidth. The authors should backup this claim with experimental measurement and a figure showing the harvested power as a function of the vibration frequency.

-From Figure 8, when the gap is reduced, the voltage is increased. What does happen if the gap is further reduced? Is there an optimal gap? Or the smaller the gap, the larger the rms voltage?

-Figure 11, what is the piezoelectric capacitance  of a piezoelectric harvester? And the frequency of the voltage? The resistance maximizing the power in case of weakly coupled piezoelectric harvesters is given by:

Ropt=1/(Cp*w)=1/(Cp*2*pi*f)

This could be verified here.

See the following references for further details and a justification of this formula:

https://journals.sagepub.com/doi/abs/10.1177/1045389X18810802  

https://iopscience.iop.org/article/10.1088/1361-665X/aaca56/meta

-Harvesting the energy on a resistance is not useful in a practical case, as the energy must be stored in a capacitor in order to power a sensor. While the reviewer understands that the authors use a resistance in order to model the input impedance of a harvesting circuit, some explanations should be added in the new version of the manuscript. It is necessary to clarify that in a practical case, a self-powered low-power electronic interface with a MPPT algorithm such as the one proposed in [A] and [B] should be implemented.

[A] A. Morel et al., "Fast-Convergence Self-Adjusting SECE Circuit With Tunable Short-Circuit Duration Exhibiting 368% Bandwidth Improvement," in IEEE Solid-State Circuits Letters, vol. 3, pp. 222-225, 2020, doi: 10.1109/LSSC.2020.3012340.

[B] Y. Cai and Y. Manoli, "A piezoelectric energy-harvesting interface circuit with fully autonomous conjugate impedance matching, 156% extended bandwidth, and 0.38μW power consumption," 2018 IEEE International Solid - State Circuits Conference - (ISSCC), 2018, pp. 148-150, doi: 10.1109/ISSCC.2018.8310227.

Minor comments

-Please do rephrase the sentences starting with “And” (such as the one in the abstract: “And corresponding theoretical analyses and experimental verifications were carried (…).”) as it is too informal.

-I advise the authors to carefully proofread the manuscripts in order to remove grammar and English mistakes.

-Some sentences are way too long. For example, in the introduction: “For example, Fu et al.[25] integrated up-conversion and a bistable mechanism by introducing two external input magnets, and this harvester has a wide bandwidth at low speed and achieves an asymmetric potential well by adjusting the magnetic potential so that the harvester has a stable and enhanced output and good control behavior, Hu et al.[26] proposed an energy recycler made of U-shaped bimorphvalues, the electrical power generated by the harvester can reach 8.19 KW, which is comparable to the traditional small windmill generator, so the proposed harvester has attractive application prospects, Xue et al.[27] proposed three configurations to achieve out-of-plane plucking: direct repulsion configuration, orthogonal configuration, and indirect repulsion configuration.”. The authors are advised to do short sentences in order to clarify the manuscript.

-Use “Figure” instead of “Fig.” when “Figure” is the first word of a sentence.

-Page 4, “Galerkin expansion” instead of “Galliakin expansion”

-Equation (17), the rms voltage should not depend on the time “t”. The integral upper bound should be “T” instead of “t” and the ratio in front of the integral should be “1/T” instead of “1/t”, with T being the period of the voltage

Round 2

Reviewer 1 Report

The author's have addressed all of this reviewer's points. Thank you very much.

Author Response

请参阅附件。

Reviewer 2 Report

The authors have satisfactorily addressed some of my concerns. However, I think that the following points still need to be addressed in order to accept the submitted paper:

-The parameters provided in Table 1 do not allow to reproduce the simulations of equations (1)-(4). For instance, the electromechanical coupling coefficient vp or the stiffnesses (K, Kg) are not provided. Could you provide in this table all the parameters that are used in equations (1)-(4), with the same notations.

-The equations (5)-(18) constitute most of the theoretical part of the paper. Have these equations been simulated or compared to experimental results?  If no, how are they useful in this paper?

-Page 12, "Meanwhilethe resistance that maximizes the power can be calculated from the following equation: R=1/(2 pi Cp f)". As explained in my previous review, the authors should add references in order to justify this sentence, as this is a well-known fact in the community.
